# Oncogene-Induced Reprogramming in Acute Lymphoblastic Leukemia: Towards Targeted Therapy of Leukemia-Initiating Cells

**DOI:** 10.3390/cancers13215511

**Published:** 2021-11-02

**Authors:** Vincent Fregona, Manon Bayet, Bastien Gerby

**Affiliations:** Centre de Recherches en Cancérologie de Toulouse (CRCT), Université de Toulouse, Institut National de la Santé et de la Recherche Médicale (INSERM), UMR-1037, Université Toulouse III Paul Sabatier (UPS), 31100 Toulouse, France; vincent.fregona@inserm.fr (V.F.); manon.bayet@inserm.fr (M.B.)

**Keywords:** oncogene-induced reprogramming, (pre-)leukemic stem cells, cell plasticity, self-renewal, oncogenic transcription factors, acute lymphoblastic leukemia

## Abstract

**Simple Summary:**

Acute lymphoblastic leukemia is a heterogeneous disease characterized by a diversity of genetic alterations, following a sophisticated and controversial organization. In this review, we present and discuss the concepts exploring the cellular, molecular and functional heterogeneity of leukemic cells. We also review the emerging evidence indicating that cell plasticity and oncogene-induced reprogramming should be considered at the biological and clinical levels as critical mechanisms for identifying and targeting leukemia-initiating cells.

**Abstract:**

Our understanding of the hierarchical structure of acute leukemia has yet to be fully translated into therapeutic approaches. Indeed, chemotherapy still has to take into account the possibility that leukemia-initiating cells may have a distinct chemosensitivity profile compared to the bulk of the tumor, and therefore are spared by the current treatment, causing the relapse of the disease. Therefore, the identification of the cell-of-origin of leukemia remains a longstanding question and an exciting challenge in cancer research of the last few decades. With a particular focus on acute lymphoblastic leukemia, we present in this review the previous and current concepts exploring the phenotypic, genetic and functional heterogeneity in patients. We also discuss the benefits of using engineered mouse models to explore the early steps of leukemia development and to identify the biological mechanisms driving the emergence of leukemia-initiating cells. Finally, we describe the major prospects for the discovery of new therapeutic strategies that specifically target their aberrant stem cell-like functions.

## 1. Introduction

Cell plasticity is a property required for cell reprogramming. The specification of cellular fate during development and differentiation is a dynamic and evolving process that initiates in stem/progenitor cells. A network of transcription factors controls the balance between the maintenance of stem cell identity and the process of lineage specification. However, somatic cell identity is not fixed and can be modified since cellular reprogramming has been achieved. Especially, the concept of the stability of stem cell function has been challenged by the finding that only the Oct4, Sox2, c-Myc and Klf4 transcription factors are required for the reprogramming of somatic cells into induced pluripotent stem cells (iPS) [1]. Interestingly, these four factors have been shown to be oncogenic in different contexts [2], suggesting a link between cell reprogramming and tumorigenesis. This observation is along the lines of the notion that cancer progression is characterized by the gradual loss of a differentiated state associated with the reprogramming of stem cell-like features. Indeed, transcriptional and epigenetic modifications occurring in malignant cells frequently lead to tumor dedifferentiation and the acquisition of stemness features [3] and can be exploited using machine learning approaches to predict the clinical outcome and to identify potential drugs targeting the stemness signature [4]. Thus, cell plasticity, oncogene-induced reprogramming, self-renewal, perturbation of lineage identity and cancer initiation appear to be tightly intertwined and should be considered as critical mechanisms for identifying and targeting the cell-of-origin of cancer.

Self-renewal is defined by the functional property of stem cell populations that undergo symmetric or asymmetric divisions to preserve tissue integrity and homeostasis in normal development. In adult hematopoiesis, sustained self-renewal is a distinctive function of normal hematopoietic stem cells (HSCs) which is tightly controlled by a network of transcription factors [5], whereas committed hematopoietic progenitors are devoid of this stem cell property. Thus, it was originally thought in myeloid malignancies that the biological features of leukemic stem cells (LSCs) come from reminiscent states of normal HSCs in which self-renewal is a central property in the process of leukemia initiation. In particular, pioneering studies in chronic myeloid leukemia (CML) considered that the primary oncogenic event takes place in rare and self-renewing multipotent stem cells [6,7], and established the concept that these cells are prone to transformation because of their unique self-renewal capacity. In acute myeloid leukemia (AML), the notion that leukemia originates from an HSC or a committed progenitor is debated in the field. The phenotypic characterization of LSCs in the original studies led to the idea that AML is organized in a hierarchical pattern, in a way similar to that of normal hematopoiesis and derive from the malignant transformation of a primitive hematopoietic cell [8,9]. Supporting this notion, some murine leukemia models have led to show that AML-initiating mutations occur at the level of HSCs and alter their self-renewal properties [10,11]. However, there is evidence that leukemia-initiating activity can be observed not only in an immature cell population but also in populations corresponding to a range of normal committed progenitors. Indeed, a number of studies brought evidence that more mature progenitors, that normally lack any potential for self-renewal, may be the cells-of-origin in AML [12,13,14,15,16]. Thus, oncogenic transcription factors can destabilize the normal molecular program of target cells, leading to changes in gene expression and to a total or partial loss of the original cell identity.

Defined as the most frequent pediatric cancer, acute lymphoblastic leukemia (ALL) is a multistep disease characterized by the acquisition of diverse genetic alterations that can be classified into more than 20 B-lineage subtypes (B-ALL) and more than 10 T-lineage subtypes (T-ALL). These different oncogenic subgroups are established according to the identity of the first oncogenic event carrying in the leukemic cells [17]. Chemotherapy is efficient at inducing long-term remission in child but is associated with severe side effects and undesirable consequences, including second malignant neoplasms. Indeed, while the 5-year survival rates now exceed 90%, the most common cause of treatment failure in pediatric ALL remains relapse that occurs in approximately 15–20% of patients [17]. In addition, the treatment is also limited by indiscriminate toxicity towards normal HSCs. Over the last decade, genome-wide analyses, mainly through next generation sequencing approaches, have been extensively used to draw the genomic landscape and the gene expression profile of ALL patients. As recently reviewed and updated both in T-ALL [18] and B-ALL [19,20], this offered major insights regarding the diversity of oncogenic subtypes associated with altered signaling pathways and led to patient stratification and to the discovery of new therapeutic opportunities. However, although current chemotherapy is efficient at reducing the tumor load by targeting proliferating and metabolically active leukemic cells, the disease relapse points to the presence of resistant cells that escape treatment [21]. Thus, pioneering works and emerging studies converge around the notion that the biological properties of pre-leukemic and/or leukemic stem cells, including oncogene-induced reprogramming, cell plasticity, sustained self-renewal activity, cell-quiescence and drug-resistance, can significantly affect leukemia treatment and should be considered in the search for new and more targeted therapies.

With a focus on B-ALL and T-ALL, we present in this review previous and current studies exploring the cellular, molecular and functional heterogeneity in leukemic cells from patients. With a particular emphasis on the lymphoid lineage, we also review the engineered mouse models that led to the recognition that a single primary oncogene could be sufficient to confer stem cell-like properties to committed progenitors, converting them to a population of pre-LSCs. Finally, we describe the perspectives for the identification of new therapeutic agents that specifically target their aberrant stem cell-like functions.

## 2. Cellular and Molecular Heterogeneity in Acute Leukemia: In Search of Leukemic Stem Cells

### 2.1. The Origin of the Leukemic Stem Cell Concept

According to several pioneering works [22,23,24,25,26,27], normal hematopoiesis is a well-known tightly regulated process based on a hierarchical organization in which a small number of multipotent stem cells maintain all of the hematopoietic lineages. Similar to the normal hematopoietic system, the concept of leukemic stem cells (LSCs) supports the idea that leukemic cells are functionally heterogeneous, following a hierarchical model in which only a minor population of LSCs residing are able to initiate and indefinitely maintain the neoplastic tissue. Historically and experimentally built on xenograft transplantations, this concept has come from pioneering studies on AML from Dick’s laboratory, which have defined a distinct subpopulation of tumor cells characterized by their capacity to initiate and propagate the disease when transplanted into immunodeficient mice [8,9]. Since this finding, xenotransplantation of human leukemic cells has become the gold standard assay to study LSC activity [28]. Based on an immunophenotypic identification, AML was thus the first hematological malignancy with a reported LSC population within the tumor bulk that sustains the long-term leukemia development. This has been confirmed by a number of subsequent studies describing a rare and specialized population of LSCs enriched in the CD34+CD38-Lineage- fraction from AML patients [29,30,31,32]. However, transplantable LSCs from AML samples could also be found in the CD34+CD38+ and CD34- subpopulations, albeit with lower frequency [28,33,34,35], and therefore could be phenotypically more diverse than originally thought. Despite this controversy, it is still inferred that in contrast to leukemic blasts that have limited self-renewal potentials, LSC-enriched populations reside at the apex of the leukemia hierarchy, are able to sustain long-term tumor growth and constitute an important driver of relapse through their slow division rate that make them resistant to conventional therapies [28,36].

### 2.2. Phenotypic and Functional Plasticity in ALL

The notion of cellular and functional heterogeneity in leukemic cells is of fundamental interest to understand the leukemia initiation and development in patients. Therefore, the hierarchical model of the LSC concept was applied to acute lymphoblastic leukemia, in both B-ALL and T-ALL. In B-ALL, scientists extensively used CD34, CD38 and CD19 markers to explore cellular and functional heterogeneity in leukemic blasts from adult and infant patients, which lead to many controversies [37]. It has originally been proposed that leukemic cells with the HSC-like immunophenotype CD34+CD19- exclusively contained LSCs in both high and standard risk B-ALL [38,39]. However, the following investigations demonstrated that self-renewal activity can be enriched in leukemic subpopulations expressing the CD19 marker [40,41,42,43], corresponding phenotypically to a range of normal B-cell precursors. Nevertheless, the prospective enrichment of LSCs in B-ALL using the CD34 and CD38 surface markers, associated or not with CD19, led to highly variable results [37,42,43,44]. These controversial conclusions could be explained by the possibility that the majority of blasts among the tumor can sustain leukemia-initiating activity, thus following a stochastic model of B-ALL. Indeed, Vormoor and colleagues challenged the hierarchical stem cell concept by demonstrating using limiting dilution assays (LDAs) that B-ALL cells able to engraft immunodeficient mice are highly frequent and are not restricted to a population with a specific immunophenotype [43,45]. Using an in vivo tracking approach by cellular barcoding of B-ALL samples combined with deep sequencing, the same group reported that leukemia-initiating clones are abundant and functionally equipotent, exhibiting a similar ability to reconstitute the disease over serial transplantations [46]. Other investigations by Rieger and colleagues demonstrated at the single-cell level that the expression of CD34 and CD38 markers is a highly plastic and dynamic process at the surface of B-ALL blasts [47], which also could explain the previous discrepancies in using these two markers for the enrichment of LSC activity. In the same line of thought, the loss of CD19 antigen expression at the surface of leukemic cells enabling the tumor to evade chimeric antigen receptor (CAR) immunotherapy [48,49] is a well-known representative example of antigen plasticity and evolved adaptation of leukemic cells upon the treatment. Therefore, the identification of several stable surface markers aberrantly expressed in leukemic cells is critical to follow the disease evolution and to monitor resistant cells [50,51]. Together, phenotypic and functional plasticity should be considered in experimental approaches to identify LSC-enriched populations, but also at the clinical level to explain therapy escape mechanisms and B-ALL relapse.

In contrast to B-ALL, there are few published studies of cell populations enriched in LSCs from human T-ALL samples. Nevertheless, original works demonstrated that leukemia-initiating activity can be explored by using in vivo xenografts and in vitro assays [52,53]. The functional heterogeneity in human T-ALL combined with the phenotypic characterization of the LSC compartment has been explored using CD34, CD7 and CD4 surface markers. In particular, the work from Pflumio’s group reported that CD34+CD7-CD4- fraction from the blood of T-ALL patients does not produce leukemic blasts but undergoes normal hematopoiesis in vitro and after transplantation into immune-deficient mice [54], which is quite similar to what has been observed for the CD34+CD19-CD38- population from B-ALL patients [41,42], and that correspond to the residual and circulating normal hematopoietic stem and progenitor cells (HSPCs). In fact, LSC activity seems to be exclusively present in the CD7+ fraction [54,55], suggesting that T-ALL initiation is triggered in a committed T-cell. Finally, T-ALL development is significantly pronounced after the xenograft of CD7+CD34+ leukemic cell population, as demonstrated in three independent studies [54,56,57]. Interestingly, despite the fact that the CD34 marker seems to be useful to enrich LSC activity from T-ALL samples, immunophenotyping of donor-derived blasts in xenograft samples revealed that CD34+ blasts may lose the CD34 maker after transplantation, and vice versa [57]. This observation suggests that CD34 could be a plastic marker at the surface of T-ALL blasts, as it has been shown at the clonal level in B-ALL [47], and asks about the functional relevance and the correspondence of LSC surface markers between blasts from patients and blasts from patient-derived xenografts (PDX).

Collectively, these studies challenge the hierarchical organization of the stem cell concept in ALL by raising the question of whether surface marker plasticity is connected to functional plasticity within the tumor. Indeed, T-ALL characteristics such as cell-surface immunophenotype, but also dormancy and chemoresistance, are under the influence of the bone marrow (BM) niches, indicating an important cellular and functional plasticity of leukemic cells in response to their in vivo microenvironment [58]. Using ALL xenograft models combined with in vivo cell-division assay, Jeremias and colleagues also addressed this notion by demonstrating that stem cell properties, such as dormancy, treatment-resistance and leukemia-initiating activity, are reversible [59]. In addition, the authors showed that the functional plasticity of dormant clones is dependent of their in vivo environment and suggest that this reversible mechanism could be involved in treatment failure and ALL relapse. The molecular characterization of those dormant and chemo-resistant clones recently led to the extraction of a core subset of genes that would help for the risk stratification of ALL patients [60], as previously achieved in AML [61]. Together, these findings raise the importance of considering cellular and functional plasticity in the clinical outcome of ALL patients. This points to the need to focus on the functional mechanisms of leukemic cells, such as self-renewal, cell-quiescence and -resistance, to identify new specific LSC markers.

### 2.3. When Molecular Diversity and Clonal Evolution Meet the LSC Theory

Accurately addressing the cell-of-origin in leukemic patients using surface markers seems to be obviously limited, not only by the plastic properties of leukemic blasts as previously described, but also by their genetic heterogeneity, which represents a major hallmark of cancers, including ALL [62,63]. Indeed, the development of acute leukemia is a multistep process characterized by the acquisition of accumulated mutations. Consistent with the observation that acute leukemias exhibit a limited number of genetic alterations [64], each mutation can perturb critical cellular functions and their combinatorial interaction would be sufficient to cause leukemia.

Through a cytogenetic approach by multiplex fluorescence in situ hybridization (M-FISH), Greaves and colleagues explored in a ground-breaking study the intraclonal genetic architecture of leukemic cells from B-ALL patients. Since the M-FISH approach allows for the detection of up to eight genetic alterations at the single-cell level, the authors provided the first direct evidence for genetic diversity of leukemic cells by establishing phylogenetic trees of clonal evolution within individual patients [65]. Major insights in the composition and the dynamic of subclones in leukemic cells have been also gained through the comparison of the genetic landscape of paired diagnostic and relapse ALL samples, as performed in numerous works [66,67,68,69,70,71,72,73,74,75]. Thus, genome-wide studies, including single nucleotide polymorphism (SNP), next generation sequencing (NGS), comparative genomic hybridization (CGH) and multiplex ligation-dependent probe amplification (MLPA) analyses of matched diagnosis-relapse ALL samples, led to an exhaustive characterization of the genetic profile of patients during the evolution of their disease. These integrative approaches helped to establish the complex architecture of individual leukemia and revealed that a relapse may be generated from major, minor or ancestral clones from the initial diagnosis. Remarkably, Mullighan and colleagues demonstrated that more than half of the relapse ALL samples lacked some of the genomic copy number abnormalities (CNAs) present at the diagnosis and acquired new and distinct genetic lesions [66,75,76]. This observation indicated that in the majority of ALL cases, relapse leukemic cells have evolved not from the bulk of the diagnosis cells but from a clone that harbored a restricted number of genetic alterations, such as a founding chromosomal translocation. Indeed, while a relapse may be produced from a predominant clone at the diagnosis, the majority of them arise from pre-existing minor subclones highly diluted within the diagnosis sample or from the clonal evolution of an ancestral clone [66,75,77,78].

By the same way of evidence, whole- and targeted-exome sequencing of peripheral blood cells from a large cohort of healthy individuals identified that 10 to 20% of people aged over 70 harbor pre-leukemic mutations resulting in the dominance of a small number of HSC-derived clones, a process called age-related clonal hematopoiesis (ARCH) [79,80]. While these pre-leukemic HSCs still have the capacity to differentiate and produce healthy blood cells, additional mutations can lead to disease progression towards myeloid malignancies such as AML [81]. In addition, ultra-sensitive deep sequencing of targeted genomic regions from AML patients revealed the existence of long-lasting clones carrying pre-neoplastic mutations, referred to as pre-leukemic HSCs hidden within the bulk of the tumor and that serve as a reservoir for disease progression [82,83,84]. These cells can thus acquire stem cell-like drug resistance mechanisms and by way of consequences, are spared by current treatment and are involved in the disease evolution and relapse. Despite of their low abundance in patients, it has been recently shown using clonal tracking from single-cell transcriptomics that these pre-LSCs exhibit a specific gene expression profile, distinct from that of leukemic cells and of normal HSCs [85].

Since xenotransplantation remains the gold standard assay to evaluate leukemia-initiating activity, the notion that genetic subclones are frequently selected in xenograft models adds a level of complexity to the LSC definition. Nevertheless, tracking leukemic clones in immunodeficient mice not only provides insights into the genetic and clonal architecture of human ALL, but also can be used to isolate resistant and adaptative subclones that participate to disease progression. Indeed, it has been shown that xenograft models of T-ALL can recapitulate at the genetic and functional levels to the gain of malignancy observed at relapse of the disease, including aggressiveness and therapy resistance [86]. Consistent with a branched rather than linear evolution, oncogenic trees showed that xenograft and relapse samples had frequently derived from an ancestral pre-leukemic clone, and not from diagnosis cells. Interestingly, in the case of *ETV6-RUNX1*-induced B-ALL, the putative ancestral pre-leukemic clone harboring the *ETV6-RUNX1* fusion only observed at the time of diagnosis does not regenerate neither over serial transplantations nor at the relapse [65]. This finding suggests that although xenografting leukemic cells allow us to explore the genetic progression and the aggressiveness of the disease, it does not represent a relevant approach to expand and isolate ancestral pre-leukemic clones. The branching and multi-clonal evolution model of leukemogenesis has been similarly explored using xenotransplantation approaches in *BCR-ABL1* [87] and in *MLL*-rearranged [88] ALL. Therefore, although debated in the literature [89,90], genetic and clonal variegations should definitely be considered to define LSCs by using transplantation in immunodeficient mice. Through an unprecedented large-scale LDA xenografting approach combined with targeted sequencing, this point was recently clarified by the isolation and the characterization of subclones from diagnosis B-ALL samples responsible for a relapse [91]. These clones, referred to as diagnosis Relapse-Initiating (dRI) clones, can be revealed only in a minor proportion of xenografted mice and display low sensitivity to chemotherapeutic agents. At the molecular level, dRI clones activate the mitochondria metabolism and unfolded protein response (UPR) [91], two stress molecular pathways that have been described to be critical in the maintenance of stem cell homeostasis and function [92,93].

A better understanding about the dynamic and the selection of leukemic cells during disease progression has also been gained through single-cell developmental classification methods. In particular, Nolan and colleagues developed a single-cell mass cytometry (CyTOF) approach [94] allowing for the simultaneous quantification of up to 35 proteins, including surface markers and intracellular phosphorylated proteins involved in normal and pathological B-cell development [95,96]. Using this strategy, they were able to align human B-cell subpopulations into a unified trajectory and to precise their regulatory signaling pathways during early differentiation checkpoints [96]. Built on this expertise and combined to machine learning approaches, they recently established a single-cell classification of human B-ALL at the diagnosis and identified specific features, such as activated mTOR signaling and unresponsive pre-BCR signaling, which were sufficient to predict patient relapse [95]. In addition, Müschen and colleagues recently used single-cell amplicon sequencing combined with a single-cell phosphoprotein analysis to study the interaction of oncogenic lesions in STAT5 and ERK signaling pathways during normal B-cell development and malignant transformation [97]. Using these approaches, the authors showed that the driver mutations in these two pathways are mutually exclusive in human B-ALL, consistent with the segregation of STAT5 and ERK phosphorylation. Moreover, the following functional experiments showed that concurrent oncogenic STAT5 and ERK activation can subvert leukemia development, demonstrating the proof of concept that the reactivation of divergent and conflicting signaling pathways represents a powerful barrier to transformation [97]. Recently, single-cell amplicon sequencing has also been applied in human T-ALL to study their clonal heterogeneity and evolution [98]. Strikingly, this approach allowed for the detection of clinically relevant subclones at diagnosis that evolved to major clones at later disease stages.

Collectively, these studies uncovered a considerable and interconnected subclonal diversity in leukemic cells from ALL patients, resulting from a complex, nonlinear and branching evolutionary pathway. They have also demonstrated that the majority of ALL relapses after chemotherapy arise from persisting and resistant minor clones already existing at the time of diagnosis, which can be revealed using xenograft models. These studies predicted the existence of an ancestral pre-diagnostic clone harboring a minimal set of genetic alterations which is not yet transformed but at the top of the tumoral hierarchy and heterogeneity. In addition to genomic approaches, this set of publications also highlights the relevance in using single-cell phosphoprotein analyses to define a precise trajectory of normal and pathological B-cell development and to anticipate the relapse of the disease.

## 3. Oncogene-Induced Reprogramming in ALL: Tracking (Pre-)Leukemic Stem Cells

### 3.1. Self-Renewal Property and Lineage Plasticity: Early Events of Leukemogenesis?

Major insights about the multistep process and the prenatal origin of ALL development have been gained from 25 years of outstanding studies by Greaves and his group, exploring leukemia initiation and progression in monochorionic twins [99]. These studies revealed that chromosomal translocations, in particular *ETV6-RUNX1*, could be detected in blood cells many years before the leukemia onset, establishing a pre-leukemic sub-clonal compartment. Thus, these pre-leukemic cells contain the founding genetic alteration but do not have the capacity to induce the disease by themselves. Later in life, the acquisition of full complement mutations transforms them into malignant leukemic cells, subject to clonal evolution and selection processes, and leads to the genetic heterogeneity of the tumor bulk at the time of diagnosis [99]. Therefore, it seems to be clear that *ETV6-RUNX1* originates prenatally during fetal hematopoiesis and acts as a first oncogenic event in a committed B-cell to induce the emergence of a pre-leukemic clone with altered self-renewal and survival properties [100]. Interestingly, the self-renewing pre-leukemic clone exhibits the combination of surface markers CD34+CD38-CD19+CD10- that corresponds neither to a normal B-cell subpopulation nor to fully transformed leukemic blasts. Together, these findings support the view that self-renewal is an early and obligatory event in leukemia initiation, a specific feature of the cell-of-origin, and differs from the propagating activity of fully transformed leukemic blasts. Obviously, the major limit is that pre-leukemic clones are very infrequent in ALL patients [65,75]. Therefore, except in rare cases of paired leukemic and pre-leukemic monochorionic twins [99], exploring their biological properties remains highly challenging.

To tackle this issue, the development of engineered mouse models in which there are activated particular oncogenes in specific lineages is of critical interest. Indeed, transgenic mice represent a valuable tool to understand the biological mechanisms by which a primary oncogene induces the disease and their use opens new challenges and has several perspectives: (i) it allows us to decipher biological mechanisms by which an oncogenic pathway perturbs normal hematopoietic development and reprograms a committed progenitor into a pre-LSC during the disease initiation; (ii) it helps to understand the multistep process of the disease from the pre-leukemic to the leukemic stages by identifying the oncogenic collaborative events driving malignant transformation; (iii) it allows us to explore at the molecular and functional levels the crosstalk between pre-leukemic and leukemic cells with their microenvironment; and (iv) finally, it aims to develop therapies that specifically target markers and/or biological mechanisms involved in leukemia development and resistance (Figure 1).

The use of AML mouse models corroborated that the genetic alterations of transcription factors are early events in leukemia development and can interfere with essential cellular functions of somatic cells. As examples, oncogenes such as *MOZ-TIF2* [12], *MLL-AF9* [13], *MLL-ENL* [14] or *PML-RAR* [15,16] were shown to be able to induce AML development when introduced into committed target cells. Precisely, the molecular and cellular characterization of the cell-of-origin from the AML mouse models led to the recognition that these cells can self-renew and exhibit a self-renewal gene signature induced by such oncogenic transcription factors. For example, *AML1-ETO* and *MLL-AF9* disrupt the normal hematopoietic functions by inducing self-renewal activity of myeloid progenitors prior the progression towards overt leukemia [13,101,102]. In the case of *MLL-AF9*, gene expression arrays have revealed that the oncogene can activate an HSC-like program in committed granulocyte–macrophage progenitors [13,102]. However, other oncogenes such as *BCR-ABLp190* are unable to confer self-renewal properties to hematopoietic progenitor cells [12]. In these cases, the self-renewal function must be conferred by the targeted cell or by additional genetic alterations. On the other hand, the abrogation of transcription factor activity could also result in impaired differentiation and the development of hematopoietic malignancies, such as myelodysplastic syndrome (MDS) and AML. For example, the loss of function mutations of the two important transcription factors RUNX1 and CEBPα have been identified in rare familial hematopoietic disorders involving a predisposition to MDS and AML. Furthermore, murine models with diminished expression of key transcription factors such as JunB, Gata1 and Gata2 have shown to perturb HSC regulation and function, establishing a pre-leukemic state, primed to undergo subsequent AML transformation [103,104,105]. These observations support the view that the loss of function of key transcription factors can also lead to the emergence of an aberrant pre-leukemic stem cell population prior to clonal transformation.

Considering the importance of self-renewal in leukemia initiation, an outstanding question remains how one oncogenic transcription factor can modify or reprogram stem cell-like properties in normal cells and can lead to the emergence of a pre-neoplastic population (Figure 1). The occurrence of biphenotypic leukemias and the concept of lineage infidelity in acute leukemias have long been thought of as a degree of plasticity in leukemic cells which either is conferred by the cell-of-origin, such as a pluripotent stem/progenitor, or results from the oncogene subversion of the process of lineage determination in a committed progenitor [106]. This long-lasting idea revives the question of to what extent oncogenic transcription factors in acute leukemias lead to the reactivation of self-renewal genes with or without pluripotency. It also asks to what extent oncogenic alterations that arise in committed progenitors lead to a whole or partial reprogramming of a normal cellular fate and open a new pathologic developmental program. Moreover, the exploration of the cell-of-origin in mixed phenotype acute leukemia (MPAL) led to the idea that a founding alteration, rather than the secondary events, primes pre-leukemic clones for lineage plasticity [107].

### 3.2. Pre-Leukemic Thymocyte Self-Renewal: Lesson from bHLH Transcription Factors

In ALL, the demonstration that an oncogenic transcription factor can reprogram a committed progenitor into an aberrant self-renewing pre-LSC has been made for the first time in thymocytes. Since bone marrow-derived progenitors settle in the thymus and gradually lose stemness properties and acquire T-cell characteristics, normal thymocytes have a very limited self-renewal capability [108]. Indeed, thymic output requires continuous seeding from HSCs-derived progenitors, and thymic progenitors progress into the thymus through several stages of differentiation (DN1-4, ISP8, DP) before giving rise to CD4^+^ or CD8^+^ immunocompetent cells [109,110]. Therefore, the thymus represents a relevant cellular platform to study oncogene-induced reprogramming. As assessed by serial transplantation of pre-leukemic thymocytes, McCormack and colleagues reported that the overexpression of the *Lmo2* oncogene in the thymus induces the emergence of a population of pre-LSCs [111]. Precisely, using Cd2-Lmo2 transgenic mouse model that recapitulate the human T-ALL, the study showed that enforced expression of Lmo2 converts normal DN3 thymocytes into a self-renewing, pre-leukemic population by reprogramming a stem cell-like gene program.

LMO proteins, members of the LIM-domain only family, lack the DNA-binding ability and require a protein interaction with basic helix-loop-helix (bHLH) transcription factors such as SCL/TAL1 (SCL) or LYL1 to form a multiprotein complex on DNA and activate the transcription that controls HSPC functions [112]. Based on this molecular evidence, protein–protein interaction should occur between bHLH and LMO1/2 oncoproteins to activate the aberrant stem cell gene program in pre-LSCs. Indeed, Hoang and colleagues showed that the ectopic expression, the interaction and the collaboration of SCL and LMO1 oncoproteins are critical to alter thymocyte differentiation [113] and to induce a self-renewal molecular network for the emergence of pre-LSCs in the thymus [114,115]. Therefore, the capacity of SCL-LMO1 to reprogram DN3 thymocytes into self-renewing pre-LSCs mirrors in some aspect the physiological function of SCL, which controls the repopulation ability and the maintenance of HSPCs [116,117,118]. Furthermore, associated with LMO1, LYL1 ectopic expression mimics the effect of SCL to activate the self-renewal function in pre-LSCs [114], corroborating their functional redundancy described in normal HSCs [119]. In contrast, the loss of function approaches showed that Lyl1, but not Scl, is required for *Lmo2*-induced thymocyte self-renewal [120]. This observation suggests that Lmo2 oncoprotein interacts preferentially with endogenous Lyl1 than with Scl for thymocyte reprogramming. According to the importance of *Scl* gene dosage in normal HSPCs [117,118], the alternative explanation would be that the physiological levels of Scl in thymocyte progenitors are not sufficient for its efficient interaction with Lmo2 oncoprotein.

Cell purification indicates that only DN3 thymocytes in both *Lmo2* or *SCL-LMO1* mouse models are able to colonize the thymus of recipient mice in transplantation assays [111,114]. The intriguing question then is what predisposes DN3 subsets to reprogramming by these primary oncogenes? The DN3 stage represents a critical restriction point in the thymus when thymocytes are committed to the T-lineage and lose non-T potential under the influence of NOTCH and pre-TCR signaling pathways. Reprogramming self-renewal activity likely occurs just prior to pre-TCR signaling since self-renewal can be induced in *Cd3*ε^−/−^ DN3a cells [114]. Thus, while pre-TCR signaling represents a collaborating event in T-ALL progression and transformation [121], it is dispensable for the initial transition from DN3 thymocytes to pre-LSCs. Therefore, NOTCH signaling, which is activated at highest levels in DN3 cells [122], represents a strong candidate to collaborate with *Lmo2* or *SCL-LMO1* oncogenes in thymocyte reprogramming. NOTCH signaling is essential for T-cell commitment and specification [123] but its role in stem cell self-renewal is controversial. Indeed, some studies indicate that NOTCH does not exert an essential function in adult HSCs [124,125,126], while others suggest that NOTCH activation enhances their self-renewal capacity [127,128]. The frequent occurrence of activating mutations of the *NOTCH1* gene in T-ALL patients [129] and in mouse models of the disease [110,130], as well as the sensitivity of both human [52,56] and murine leukemic blasts [131] to NOTCH1 inhibitors, strongly indicates that a gain of function of NOTCH1 represents a critical step in cell transformation. Nonetheless, enforced expression of NOTCH signaling by itself does not convert thymocyte progenitors into pre-LSCs [111,114], comforting the notion that *NOTCH1* mutations observed in patients are collaborating and not initiating events in the leukemogenesis process. However, pre-LSC self-renewal induced by *SCL-LMO1* collaborates with the physiological levels of the NOTCH-MYC signaling axis [114], which is provided by the cellular interactions with thymic stromal cells. Thus, in contrast to leukemia propagating cells which have acquired different sets of secondary mutations, pre-LSCs are believed to be genetically and phenotypically stable, are still capable of differentiating into mature and functional T-cells and remain highly dependent on their thymic microenvironment [111,114,121].

The major advantage of the transgenic models is to provide unrestricted and reproducible access to pre-LSCs that allow us to study and target their cellular and molecular mechanisms and particularly those involved in therapy resistance. Recent works showed that pre-LSCs are resistant to chemotherapeutic agents because of their distinctive slow-division rate [132,133]. Indeed, it is well known that cell quiescence may be an important limitation for therapeutic efficiency, as exemplified at the clinical level in CML [134,135]. Using doxycycline-inducible *H2B-GFP*^tg^ mice, a gold standard in vivo model to study cell quiescence of normal HSCs [136,137], Curtis and colleagues demonstrated that self-renewal, drug resistance and clonal evolution are restricted to a rare and slow-cycling population of pre-LSCs in the *Lmo2*-induced T-ALL model [133]. This work not only defines for the first time the importance of the cell cycle restriction in pre-LSC activities, but also makes the *H2B-GFP* system as a powerful in vivo tool to purify slow-cycling pre-LSCs in other models and to develop strategies targeting the quiescence and the resistance of relapse-inducing clones.

Collectively, this set of publications provided new insights into the long-lasting questions concerning the cell-of-origin of T-ALL and the molecular mechanisms by which oncogenic transcription factors can reprogram the self-renewal property to thymocytes, thus establishing a pre-leukemic state before malignant transformation. These studies demonstrate that the main effect of the transcription factor SCL, associated with its partner LMO1/2, is to induce DN3 thymocyte self-renewal. While thymocyte reprogramming induced by other T-ALL transcription factors such as TLX1, TLX3 or HOXA remains unexplored, emerging studies indicate that chromosomal rearrangements generating NUP98 fusion proteins can induce thymocyte self-renewal prior the DN3 stage [138,139]. These observations suggest that only particular cells possess the necessary molecular background to allow for oncogene reprogramming, and conversely, only some oncogenes, in the right cellular context, can induce stem cell-like properties. Despite this, the question of whether a primary oncogene in ALL induces sufficient molecular and functional plasticity to cause lineage subversion remains poorly explored. Interestingly, SCL and LMO2 belong to the five transcription factors that convert adult fibroblasts to multipotent hematopoietic progenitors [140], suggesting their critical functions in cellular and molecular reprogramming of committed cells. Using the generation of targeted mouse lines conditionally expressing *Lmo2*, two recent studies demonstrate that *Lmo2* acts as a “hit-and-run” oncogene in T-ALL development [141,142]. Precisely, while pre-LSC self-renewal activity required continuous Lmo2 expression, overt leukemia frequently evolves in a Lmo2-independent manner [142]. This observation strongly suggests that oncogenic evolution occurring in fully transformed blasts overcomes the requirement of the initiating event. In addition, B-cell-restricted expression of Lmo2 reprograms committed B-cells into malignant T-ALL [141], demonstrating the proof of concept that a primary oncogene can induce sufficient reprogramming to switch from a B-cell fate to a T-cell neoplastic process. The cell-of-origin and the role of the primary oncogene *ETV6-RUNX1* in lineage organization have also been addressed recently [143]. Using a lineage-specific oncogenic activation approach, Vicente-Duenas and colleagues demonstrated that *ETV6-RUNX1* can give rise to both B-ALL and T-ALL and that leukemia transformation is dependent from the nature of the acquired secondary mutations [143]. Although *ETV6-RUNX1* is always associated with B-ALL in humans, this strategy predicts that pre-leukemic clones exhibit a T-cell potential. Together, these findings have important implications to develop targeted therapies of pre-LSCs.

### 3.3. Cell Reprogramming by PAX5 Mutants: To B-Cells or Not to B-Cells?

Adult B-cell development is initiated in the BM by the entry of hematopoietic progenitors into the B-cell lineage transcription program and the concomitant sequential rearrangements of the immunoglobulin genes through V(D)J recombination, leading to the generation of immune-competent plasma cells. B-cell development can be dissected into pre-pro-B, pro-B, pre-B, immature B and mature-B cell populations corresponding to different stages of differentiation [144]. B-lineage commitment is characterized by the rearrangement of the immunoglobulin heavy-chain (*IgH*) locus that occurs during the differentiation of pre-pro-B cells to pro-B cells. Productive V_H_-D_H_-J_H_ recombination leads to the expression of the Igμ protein as part of the pre-B cell receptor (pre-BCR), which promotes the transition from the pro-B to the pre-B cell stage [145]. Then, successful immunoglobulin light-chain (*IgL*) gene rearrangement in pre-B cells results in the emergence of immature IgM^+^ B cells that emigrate from the BM to peripheral lymphoid organs [146]. B-cell lineage is generated from HSCs by a complex and tightly regulated differentiation processes. Among them, the *PAX5* gene encodes the crucial transcription factor PAX5, which has been described as the guardian of B-cell identity [147,148]. Indeed, while the developmental progression from early B progenitors to plasma cells is regulated by several extracellular signals and transcription factors [149], PAX5 is considered as a master piece to control the irreversible commitment of lymphoid progenitors to the B-cell lineage by activating the transcription of B-cell-specific programs and by suppressing alternative lineage choices [147,150,151]. The initial work demonstrated that B-cell differentiation is completely blocked at the transition between pre-pro-B and pro-B stages in *Pax5*^−/−^ mice, revealing the critical role of Pax5 in the B-cell commitment [152]. Then, functional experiments, including transplantation assays, revealed that *Pax5*^−/−^ B-cells had the ability to home in on the BM, where they exhibited aberrant engraftment activity [153]. In addition, uncommitted *Pax5*^−/−^ B-cells are able to differentiate into several hematopoietic cell types, either in vitro in the presence of appropriate cytokines or in vivo after transplantation [147,153,154,155]. Finally, Pax5 inactivation in mature B-cells leads to the downregulation of B-cell-specific genes and the reactivation of lineage-inappropriate genes [156,157,158]. This process allowed for their conversion into functional T cells exhibiting *IgH* and *IgL* gene rearrangements by dedifferentiation back to early uncommitted progenitors [159]. Interestingly, the loss of lineage identity by the elimination of Pax5 in mature B-cells is also associated with an increased risk of lymphoma development [159]. Together, these findings highlight the critical role of Pax5 in B-lineage commitment and in maintaining the B-cell identity. Given the importance of normal Pax5 in the B-cell development and the degree of plasticity of committed B-cells, the question of whether and to what extent genetic alterations involving *PAX5* could reprogram B-cell progenitors and perturb their identity and function remain to be fully explored.

*PAX5* is a well-known haploinsufficient tumor suppressor gene in human B-ALL. Indeed, heterozygous deletions of *PAX5* are found in about one-third of patients [160,161,162]. These alterations, which are considered as secondary events in B-ALL development, result in the reduction of PAX5 expression or impairment of DNA-binding activity and/or transcriptional activity of PAX5. However, *Pax5* heterozygous mice exhibit normal B-cell development and is not sufficient to induce leukemia in the absence of other oncogenic lesions [163]. The tumor suppressor role of Pax5 has been revealed in these mice using chemical-induced mutagenesis approaches [164]. In addition, the Pax5 tumor-suppressor activity has been demonstrated through the modulation of other critical transcription factors, such as Ebf1 [165,166], Ikzf1 [167] and Stat5 [168]. Since genetic lesions of *PAX5*, *EBF1* and *IKZF1* are commonly found in human B-ALL, it suggests that the dosage of these three genes is critical to prevent the disease [169,170]. Interestingly, combined heterozygous deletion of *Pax5* and *Ebf1* partially blocks the differentiation of pro-B cells and increases their lineage plasticity before the leukemia onset [165,166,171]. Indeed, *Pax5*^+/−^*Ebf1*^+/−^ pro-B cells downregulated critical genes for the preservation and the stability of the B-cell identity, revealing their response to NOTCH signaling and their T-cell potential in vitro and in vivo. Although the authors did not detect any sign of dedifferentiation into classical hematopoietic progenitors, they demonstrated in this study that simultaneous reduction of Pax5 and Ebf1 induces at the molecular and functional levels a process of lineage conversion in pre-leukemic pro-B cells [171]. Interestingly, molecular reprogramming has been also observed in *Ikzf1* [172] deficient mouse models before the leukemia onset. Thus, these studies reinforce the notion that reprogramming, a partial loss of B-cell identity and leukemia initiation are critical mechanisms involved in pre-leukemic cells prior to malignant transformation. Interestingly, recent works from Sanchez-Garcia and colleagues demonstrated that leukemia initiation and progression can be triggered upon natural infection exposure [173,174] or by altering the gut microbiome [175] in *Pax5* heterozygous mice as well as in the *Sca1-ETV6-RUNX1* model. Indeed, the production of inflammatory cytokines triggered by infection can promote leukemia growth in *Pax5*^+/−^ mice [176]. These studies strongly support the epidemiological evidence suggesting that patterns of infection and inflammation after birth have a causal role in triggering the acquisition of full complement mutations in pre-neoplastic cells to transform them into malignant leukemia [99]. Indeed, emerging evidence demonstrates that a major driver for the conversion from a pre-leukemic clone to fully transformed B-ALL is the exposure to immune stressors, which currently represent a major therapeutic opportunity in the field [177].

*PAX5* is also rearranged in 2.6% of pediatric B-ALL cases, being fused to a wide diversity of fusion partners involving other transcription factors such as ETV6 and FOXP1, chromatin regulators such as NCoR1 and BRD1, a protein kinase such as JAK2 and an estrogen-related receptor such as ESRRB [160,178,179,180,181]. All PAX5 fusion proteins conserve the N-terminal DNA-binding paired domain of PAX5 and lack the C-terminal domain, including the transactivation domain [160,180]. Chromosomal translocations involving *PAX5* have been associated with a blockage of B-cell differentiation, the first reported and frequent example being *PAX5-ETV6*, which fuses the PAX5 paired domain to almost the entire ETV6 transcription factor [182]. In contrast to *PAX5* deletions, *PAX5* translocations act as primary oncogenic events, altering normal B-cell development in the early steps of the disease. Therefore, PAX5 fusion proteins represent strong candidates to reprogram stem cell features in B-cell progenitors. However, studying their role in leukemia development is highly limited by the lack of animal models. Exhaustively, transgenic mice expressing PAX5-ETV6 and PAX5-FOXP1 fusion proteins from the *Pax5* locus have been generated. While they failed to induce B-ALL on their own, both fusion proteins blocked the B-cell development at the pro-B/pre-B cell transition [183]. Recently, our group developed a new genetically engineered mouse model of B-ALL induced by PAX5-ELN [184], a fusion protein previously identified in patients [179]. Driven by the *IgH* locus, PAX5-ELN efficiently induced the B-ALL in mice associated with the acquisition of secondary mutations in genes involved in the JAK/STAT and RAS/MAPK pathways, which are recurrently found in B-ALL patients [162] and in other oncogene-induced B-ALL models such as *ETV6-RUNX1* [185] and *TCF3-PBX1* [186] transgenic mice. Thus, these transgenic mice accurately reproduce the key features of leukemia development and provide a valuable in vivo model mimicking the multistep process of human B-ALL. Importantly, the leukemia development is preceded by three months of a pre-leukemic phase which is also associated with a partial blockade of differentiation at the transition of pro-B/pre-B stages. Thus, our in vivo model opens the way for deciphering the biological mechanisms by which a PAX5 fusion protein reprograms normal B-cell progenitors, perturbs B-cell identity and leads to the emergence of pre-LSCs. In addition, it should help us to explore the molecular and the functional crosstalk between pre-LSCs and their BM niches.

Point mutations of *PAX5* are found in about 7% of both adult and childhood B-ALL [161,162]. Somatic mutations of *PAX5* are a hallmark of B-ALL [161,162] and inherited *PAX5* mutations have also been reported [187,188]. In contrast to PAX5 translocations that lead to the loss of trans-activating/inhibitory domains and conserves the integrity of the DNA-binding domain, *PAX5* point mutations are predicted to result in lost or altered DNA-binding or transcriptional regulatory functions [162,189]. The somatic mutation *P80R* of *PAX5* (*PAX5-P80R*) was recurrently identified within the exon 3 encoding the paired DNA-binding domain and represents the most frequent *PAX5* point mutation [161,162]. Recently, it has been shown that the *PAX5-P80R* mutation induces a unique transcriptional program in patients, defining an independent B-ALL subtype and supporting the notion that *PAX5-P80R* is an initiating lesion in the process of leukemogenesis [190,191,192,193]. However, the question of whether *PAX5* mutations and PAX5 fusion proteins exhibit different molecular mechanisms in driving B-ALL remains unanswered. Therefore, comparing the effect of PAX5 fusion proteins and *PAX5* mutations on B-cell reprograming and B-cell identity should bring novel insight on the general biological mechanisms required for normal and pathological B-cell development.

## 4. Murine Models as Tools to Explore the Multistep Process of ALL

As documented in the previous sections, murine models represent valuable tools to explore oncogene-induced reprogramming that occurs in the early step of leukemia development. Furthermore, their use has greatly contributed to understanding the natural history of the human disease from the pre-leukemic state to the following fully transformed stages. Here, we review and update the mouse models that have been developed for mimicking the multistep process of leukemia development both in T-ALL and B-ALL. Thus, we summarize the mouse strains that have been used to study the impact of the inappropriate expression of an oncogenic transcription factor on pre-leukemic T- or B-cell lineages, and to identify their associated collaborative events driving malignant transformation (Table 1).

## 5. Targeting Cell- and Non-Cell-Autonomous Properties of (Pre-)LSCs

The search for more targeted therapies remains an important challenge based on an understanding of both cell-autonomous and non-cell-autonomous mechanisms involved in leukemia initiation and propagation. Targeting genes encoding for key players in controlling proliferation and differentiation in leukemia remains a dream and, unfortunately, represents very few examples of success in the clinic so far. In addition, most transcription factors, which exert intrinsic functions on cancer cells, are considered “undruggable“ by small molecules or blocking antibodies due to their structural configurations, their protein–protein or protein–DNA interactions and their cellular localizations [209,210]. Despite this, the treatment of acute promyelocytic leukemia (APL) represents a prime example of a drug combination that targets both differentiation blockade and the self-renewal activity of LSCs through the proteasome-dependent degradation of the PML-RARα fusion protein, as described in the pioneer work from de The and colleagues [211,212]. This strategy underscores the importance of targeting the oncoprotein functions as this allowed for a more than 90% cure rate and very rare relapses, representing a major success in the field of hematologic malignancies [213]. In addition, this demonstrates the proof of concept that targeted degradation of a driver oncogenic transcription factor, along with its associated self-renewal and stem cell-like functions, is the strategy of choice for the long-term cure of leukemia. In ALL, while deciphering the diversity of the different oncogenic subtypes associated with their altered signaling pathways led to the discovery of new therapeutic opportunities [18,19], a significant proportion of patients still have unsatisfactory outcomes. Thus, the importance of challenging therapeutic approaches and of finding new druggable targets that enhance anti-leukemic efficiency remains a topical issue. In that purpose and as a representative example of targeted therapy, it has been demonstrated that the demethylase activity of UTX is essential for the maintenance of SCL/TAL1-positive T-ALL and not for other oncogenic subtypes [214]. Consistently, pharmacological inhibition of UTX with the H3K27 demethylase inhibitor GSK-J4 selectively targets SCL/TAL1-positive leukemic cells. Based on this targeted epigenetic vulnerability, the authors proposed for the first time a subtype-specific therapy in T-ALL [214].

The current cancer therapies target proliferating and metabolically active cancer cells and efficiently reduce the tumor load. However, frequent relapses indicate the persistence of residual resistant cells that escape treatment likely due to a protective microenvironment that blunt their chemosensitivity [215,216,217,218,219]. Indeed, specific locations within the niche may modify the chemosensitivity of primary leukemic blasts [58,220]. In addition, growing evidence supports the notion that leukemic cells can remodel their niche into an abnormal environment that contributes to neoplastic progression at the expense of normal hematopoiesis [221]. Thus, the interactions between leukemic cells with the molecular cues provided by their microenvironment are critical to promote leukemia progression and to design new targeted therapies, as is well documented elsewhere both in T-ALL [222,223] and in B-ALL [224]. Interestingly, non-cell-autonomous mechanisms can control the vulnerability of leukemic cells upon treatment, as exemplified by mitochondria transfer from activated stromal cells to rescue ALL cells from drug-induced oxidative stress [225]. Non-cell-autonomous pathways can also be exploited to develop therapeutic opportunities, as recently documented by Van Vlierberghe and colleagues showing that endogenous IL7, a cytokine abundantly secreted by the microenvironment, collaborates with glucocorticoids to reveal the sensitivity of residual leukemic cells upon the PIM inhibitor [226,227]. Together, the impact of extrinsic signals in leukemic cells, especially the molecular crosstalk between (pre-)LSCs and their microenvironment should be considered to develop new therapeutic approaches. Modifying the BM niche represents an innovative therapeutic approach to prevent leukemia re-initiation from residual (pre-)LSCs after complete remission. Thus, there is a need to characterize the adhesion molecules and soluble factors involved in the crosstalk between (pre-)LSCs and BM stromal cells. While the in vivo method stays the gold standard approach to study the molecular interactions between (pre-)LSCs and the BM microenvironment, a 3D organotypic “leukemia on-a-chip” microphysiological system has been recently developed. This ex vivo bioengineered strategy recapitulates the heterogeneity of the leukemic BM microenvironment and opens the opportunity to perform a niche-based drug screening assay [228].

As described previously using transgenic T-ALL models, a primary oncogene can introduce cell quiescence, drug resistance and self-renewal to pre-LSCs [132,133], three critical properties that can be controlled by their niches. Thus, the development of new drugs targeting specific non-cell-autonomous mechanisms in (pre-)LSCs could open new therapeutic opportunities for more selective treatments. This notion has recently been exploited through the demonstration that a potent inhibitor of dynamin GTPase activity overcome the chemoresistance of pre-LSCs by inhibiting the transduction of key signaling pathways provided by the microenvironment [229]. Therefore, the reliance of leukemic cells on their niches and on non-cell-autonomous pathways together with the failure of chemotherapy in a significant proportion of patients point to the need for novel drug screening strategies incorporating elements of the microenvironment. This has been challenged by the design of the niche-based screening in multiple myeloma (MM) to identify Food and Drug Administration (FDA)-approved compounds that overcome stroma-induced drug resistance [230]. Precisely, this strategy led to the discovery of a chemical inhibitor specifically targeting microtubule-bound kinesin-5 in MM cells and allowing a greater selectivity over normal hematopoietic cells [230]. Obviously, the major limit is to target these (pre-)LSCs in primary patient samples, which are not easily amenable to high-throughput screening (HTS) for drug discovery due to their extreme rarity. Despite this, the critical dependency of primary (pre-)LSCs for their microenvironment has been challenged by the design of miniaturized, niche-based assays for chemical screening strategies using both *MLL-AF9*-induced AML [231] and *SCL-LMO1*-induced T-ALL [132] models. The latter led to the identification of the 2-methoxyestradiol, an FDA-approved drug inhibiting both the cell-intrinsic SCL oncoprotein and the stroma-dependent NOTCH1-Myc pathway, together with an absence of toxicity towards normal HSC functions. As previously reported, *MYC* is indeed essential for NOTCH1 activity in T-ALL [232,233], and inhibiting *Myc* expression via a BRD4 inhibitor effectively killed leukemia-initiating cells [234,235].

Collectively, recapitulating tissue-like properties of primary (pre-)LSCs represents a promising avenue for developing new cancer therapies. In addition, drug repositioning seems to be a powerful alternative strategy to discover novel biological applications of existing drugs that have a well-established dose regimen with favorable pharmacokinetics and pharmacodynamics properties as well as tolerable side effects [236]. In addition, drug repositioning should significantly reduce the time and the cost of the pre-clinical trial. Therefore, the chemical screen of FDA-approved compounds represents an attractive approach to the discovery of novel biological applications of a known drug and its rapid translation to the clinic.

## 6. Conclusions

The leukemic stem cell concept has important clinical implications and is already applied for patient stratification and for the development of more effective therapies against hematopoietic malignancies. In this review, we presented multiple lines of evidence for the complexity of leukemic cells from ALL patients that include cellular, molecular, genetic and functional heterogeneity. We described the different technological approaches that led to the recognition that leukemic cells evolve towards a nonlinear evolution process during the progression of the disease. Then, we focused on the development of improved methods, particularly the use of transgenic mice, to detect, purify and characterize pre-malignant stem cells before they have acquired additional genetic alterations and propagating properties. We described how a primary oncogenic transcription factor can reprogram stem cell-like functions to normal progenitors, such as self-renewal and cell dormancy, and that led to the emergence of pre-LSCs. Finally, we discussed the biological as well as the clinical importance of studying and targeting the intrinsic and extrinsic properties of pre-LSCs.

Over the last decade, genome-wide studies greatly improved the classification of ALL patients, enabling the identification of diverse and heterogeneous oncogenic subtypes that led to a better patient stratification, clinical prognosis and treatment orientation. Despite considerable improvements in overall survival, a significant proportion of ALL patients still have unsatisfactory outcomes. Moreover, the finding of therapeutic strategies targeting both the self-renewal and differentiation blockade of leukemic cells in ALL has not yet been achieved. Based on a better understanding of the intratumoral heterogeneity and on the deciphering of both the cell-autonomous and non-cell-autonomous mechanisms driving the emergence of leukemia-initiating cells, we expect in the near future the identification of new druggable targets and therapeutic opportunities that might render ALL a curable disease.

## Figures and Tables

**Figure 1 cancers-13-05511-f001:**
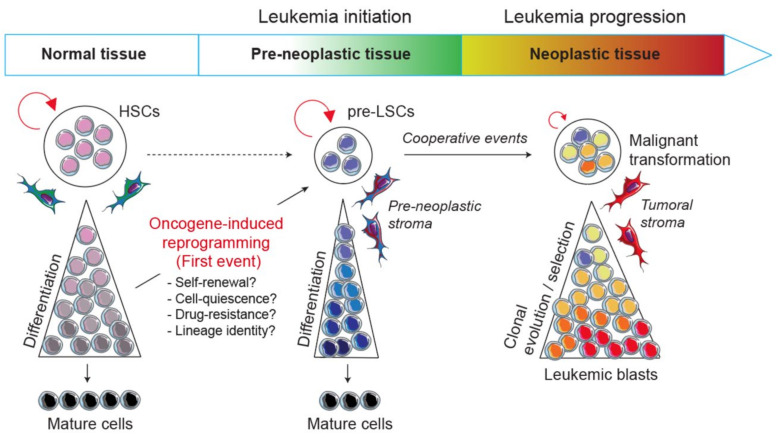
Model of leukemic evolution. A first oncogenic event can either convert a normal HSC or reprogram a normal committed progenitor into a self-renewing pre-LSC. Thus, pre-LSCs are the first cells carrying the initial pre-leukemic lesion but are still able to differentiate and give rise to mature cells. Next, following secondary and cooperative events transform pre-LSCs into malignant leukemic cells that are subjected to clonal evolution and selection processes.

**Table 1 cancers-13-05511-t001:** T-ALL and B-ALL mouse models exploring the biological mechanisms by which an oncogenic transcription factor perturbs normal hematopoietic development during the pre-leukemic phase and leads to malignant transformation.

Transgene	Additional Modifications	Pre-Leukemic State/ Reprogramming	Disease Evolution/ Collab. Events	Ref.
T-ALL mouse models				
bHLH transcription factors and partners				
Scl (*pSil, Cd2 or Sca1* promoters)	-	Perturb T-cell differentiation	No leukemia	[194,195,196]
Scl (*Lck* promoter)	-	Perturb T-cell differentiation	T-ALL (low penetrance)	[131,197,198,199]
Lmo2 (*Lck* promoter)	-	Perturb T-cell differentiation	T-ALL (low penetrance)	
Scl-Lmo2 (*Lck* promoters)	-	Expansion of DN3/DN4 populations	T-ALL (~3 months) *Notch1* mutations	
Lmo2 (*Cd2* promoter)	-	Pre-leukemic DN3 thymocyte self-renewal (pre-LSCs) ETP-ALL-like molecular signature	T-ALL (~10 months) *NOTCH1* and *Dnm2* mutations	[111,133,200]
	Scl^Δ/−^ or Lyl1^−/−^	Lyl1, but not Scl, required for pre-LSC self-renewal	Lyl1 is required for T-ALL development	[120]
	Hhex^Δ/−^ or Kit^Wv/Wv^	Hhex regulates Kit to promote radioresistance and pre-LSC self-renewal	Hhex and Kit are not required for T-ALL development	[201]
	Dnm2^V235G^	Restore cycling and survival of pre-LSCs	Accelerate T-ALL progression	[200]
	H2B-GFP (Tet-on system)	Identification of chemo-resistant and cell cycle-restricted pre-LSCs	Cell cycle restriction is critical for clonal evolution	[133]
	H2B-GFP Cdkn1a^−/−^	Loss of asymmetric cell division and cell cycle restriction	Reduce T-ALL progression	
Lmo2 (*Tet-off* system)	-	Lmo2 expression required for pre-LSC self-renewal	Lmo2-independent T-ALL development (~12 months) *Ikzf1* deletions	[142]
Lmo2 (*Cre*-driving *Rosa26* promoter)	-	Lmo2 reprograms committed B-cells into malignant T-ALL	Lmo2: “hit-and-run” oncogene in T-ALL	[141]
SCL-LMO1 (*pSil and Lck* promoters, respectively)	-	Impaired T-cell differentiation Pre-leukemic DN3a thymocyte self-renewal (pre-LSCs) Activation of a stem cell gene signature Pre-LSCs low proliferation rate and chemosensitivity	T-ALL (~4 months) *NOTCH1* mutations	[113,114,121,132]
	NOTCH1 (*Lck* promoter)	Enhance pre-LSC self-renewal Expand the pool of pre-LSC to DN1-4 and ISP8 populations	Accelerate T-ALL progression (~1 month)	[114,121]
	Cd3ε^−/−^ and NOTCH1 Cd3ε^−/−^	Pre-TCR dispensable for pre-LSC self-renewal	Pre-TCR required for T-ALL development	
LYL1-LMO1 (*Lck* promoters)	-	Pre-leukemic DN3 thymocyte self-renewal	ND	[114]
Fusion proteins				
NUP98-HOXD13 (*Vav* promoter)	-	Pre-leukemic DN2 thymocyte self-renewal (pre-LSCs) Overexpression of HOXA cluster genes	MDS/AML (mostly)/T-ALL (~15%)	[138,202]
	Lyl1^−/−^	Lyl1 required for pre-LSC self-renewal	T-ALL (100%)	
B-ALL mouse models				
Pax5 dosage				
Pax5^−/−^	-	Blockade at pre-pro-B stage (B220^+^CD19^-^Kit^+^) Self-renewal activity, multilineage potential	No leukemia	[147,153]
Pax5^Δ/−^ (*Cre*-mediated deletion)	-	Lineage plasticity and dedifferentiation of mature B-cells Conversion of B-cells into BCR-rearranged functional T-cells	B-cell lymphoma	[159]
Pax5^+/−^	-	No differentiation blockade	No leukemia	[163,203]
	Chemical (ENU) or retroviral (MMLV) mutagenesis	-	B-ALL (~6 to 8 months) *Pax5* and *Ikzf1* deletions, *Jak1* and Jak3 mutations	[164]
	Infection exposure or microbiome disturbance	-	B-ALL (~6 to 16 months) *JAK3*, *Stat5b* and *Pax5* mutations	[173,174,175]
	Ebf1^+/−^	Partial blockage at pro-B stage Reveal oncogenic potential of Il7-Myc axis Lineage plasticity of pro-B cells (T-lineage conversion)	B-ALL (~7 months) *Jak1*, *Stat5b*, *Cblb* and *Myb* mutations	[165,166,171]
	Ebf1^+/−^ Ikzf1^+/−^	-	B-ALL (40%) T-ALL (35%)	[204]
	CA-Stat5b	-	B-ALL (~2 months)	[168]
Pax5 mutations				
Pax5 P80R	- Pax5^P80R/+^	-	B-ALL (~5 months)	[190]
	-Pax5^P80R/P80R^	-	B-ALL (~3 months) *Jak1* and *Jak3* mutations	
PAX5 Y351*	-Pax5^Y351*/+^	-	B-ALL (~25 months)	[205]
	-Pax5^Y351*/Y351*^	-	B-ALL (~8 months) *Jak3* and *Ptpn11* mutations	
Pax5 rearrangements				
PAX5-ETV6 (*Pax5* promoter)	-	Blockade between pro-B/pre-B cells	B-ALL (<10%)	[183]
	Cdkn2ab^+/−^	-	B-ALL (~6 months)	
PAX5-ELN (*IgH* enhancer)	-	Partial blockage at pro-B stage Aberrant expansion potential of pre-leukemic pro-B	B-ALL (~6 months) *Ptpn11*, *Kras*, *Jak3* and *Pax5* mutations	[184]
Other rearrangements				
ETV6-RUNX1 (*Etv6* promoter)	-		No leukemia	[185]
	Sleeping beauty (SB) transposon system	-	B-ALL (~6 months) Jak1 and Jak3 mutations	
ETV6-RUNX1 (*Sca1* promoter)	-	-	No leukemia	[175,206]
	Infection exposure or microbiome disturbance	Perturb B-cell development	B-ALL (~11%)	
ETV6-RUNX1 (*Cre*-driving *Etv6* promoter)	-	Pre-leukemic *ETV6-RUNX1* HSPC (*Sca1*-Cre/Infectious stimuli) Propensity to trigger T and B malignancies	B-ALL (*Pax5* mutations) T-ALL (*NOTCH1* and *Bcl11b* mutations)	[143]
TCF3-PBX1 (*TCR V*β and *Lck* promoters /*Igμ* enhancer)	-	-	B-ALL (13%) and T-ALL (33%)	[207]
	CD3ε^−/−^	-	B-ALL (40%)	
TCF3-PBX1 (Cre-driving *TCF3* promoter)	-	Partial blockage at pro-B/pre-B stage	B-ALL (Cd19-Cre, 7%) B-ALL (Mb1-Cre, 53%) B-ALL (Mx1-Cre, 59%) *Ptpn11*, *Kras*, *Nras*, *Jak1*, *Jak3* and *Il7r* mutations *Cdkn2a* deletions	[186]
	Pax5^+/−^	-	B-ALL (increased penetrance)	
BCR-ABL1 (*Sca1* promoter)	-	Pre-leukemic *BCR-ABL1* HSPCs	B-ALL (low penetrance)	[208]
	Pax5^+/−^	Pro-B/pre-B cells permissive for B-ALL development	B-ALL (high penetrance, ~10 months)	

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
