# Peer review of "Oncogene-Induced Reprogramming in Acute Lymphoblastic Leukemia: Towards Targeted Therapy of Leukemia-Initiating Cells"

_cancers, 2021, doi:10.3390/cancers13215511_

Round 1
Reviewer 1 Report
This reviewed version of the paper has been improved according to requests of the reviewers. Editing errors have been corrected, a figure has been added for improving readability, content has been improved in several places and references have been added
Thus, to my view, the paper is now suitable for publication in Cancers
Reviewer 2 Report
Dear Sirs, I agree with all the additions and changes made.
Kind Regards
This manuscript is a resubmission of an earlier submission. The following is a list of the peer review reports and author responses from that submission.
Round 1
Reviewer 1 Report
This comprehensive review on oncogenic mechanisms in acute lymphoblastic leukemia is an interesting paper covering important biological aspects of T and B ALL.
A table presenting data obtained with transgenic mice models is adding value to the work.
I suggest to add and comment in the chapter on PAX5 the PAX5-ESRRB fusion gene that was recently described (
Indeed, this fusion involves an estrogen related receptor, which is of interest when considering the later discussed potential role of an estradiol derivate to target LSCs.
the phrase line 386 must be rewritten, since it is not understandable
"In ALL, experimental works demonstrating that an oncogenic transcription factor can reprogram non-stem cell populations into aberrant self-renewing pre-LSCs come from the thymic system."
Author Response
Response to Reviewer 1 Comments
This comprehensive review on oncogenic mechanisms in acute lymphoblastic leukemia is an interesting paper covering important biological aspects of T and B ALL.
A table presenting data obtained with transgenic mice models is adding value to the work.
Point 1: I suggest to add and comment in the chapter on PAX5 the PAX5-ESRRB fusion gene that was recently described (Haematologica Vol. 101 No. 1 (2016): January, 2016; PAX5-ESRRB is a recurrent fusion gene in B-cell precursor...). Indeed, this fusion involves an estrogen related receptor, which is of interest when considering the later discussed potential role of an estradiol derivate to target LSCs.
Response 1: We thank the reviewer for this reference that we have added in the text associated with the following sentence: “PAX5 is also rearranged in 2.6% of pediatric B-ALL cases, being fused to a wide diversity of fusion partners involving other transcription factors such as ETV6 and FOXP1, chromatin regulators such as NCoR1 and BRD1, a protein kinase such as JAK2 and an estrogen-related receptor such as ESRRB”
Point 2: The phrase line 386 must be rewritten, since it is not understandable "In ALL, experimental works demonstrating that an oncogenic transcription factor can reprogram non-stem cell populations into aberrant self-renewing pre-LSCs come from the thymic system."
Response 2: We thank the reviewer to have noticed this incomprehensible sentence that we have changed with: “In ALL, the demonstration that an oncogenic transcription factor can reprogram a committed progenitor into an aberrant self-renewing pre-LSC has been made for the first time in thymocytes.”
Reviewer 2 Report
In this work the group of Gerby provides a well written and exhaustive review exploring the concept of Leukemia Initiating Cells and their different chemosensitivity compared to bulk tumor cells.
They also provide a broad overview of the use of murine models to investigate genetic reprogramming at the inception of Leukemia development.
The concluding paragraph concerns new therapeutic strategies exploiting the processes characterizing the advent of Leukemia Initiating Cells.
Minor comments:
- The authors should proofread the review, as there are a few minor typographical and grammar errors.
- The group could find a synonym for the phrase “in the same line of thought (or evidence)” which is often reposted at lines 157, 234, 288, 381, 489, 553
- Please check that acronyms are specified when first appearing in the paper. For instance in line 176 Hematopoietic Stem and Progenitor Cells (HSPCs)
- In line 473 there is the URL of the paper “NUP98-PHF23 and NUP98-HOXD13 oncogenes confer aberrant self-renewal potential to thymocyte progenitors”. If the authors plan to use it as reference they have to add it as ref. 137 and then renumber all subsequent references.
- Authors should check that the symbols of genes and proteins are respectively in italics and in capital letters.
- I would suggest, if possible, to add a couple of figures. For example:
- A figure regarding relapsed leukemic cells evolved from clones with genetic alterations and not from bulk cells
- A figure about the concept of self-renewal in leukemic initiation
- A figure that displays the role of bHLH and LMO1/2 in pre-LSCs activation
- A figure concerning the role of PAX5 in normal and abnormal B-cell development
Author Response
Response to Reviewer 2 Comments
In this work the group of Gerby provides a well written and exhaustive review exploring the concept of Leukemia Initiating Cells and their different chemosensitivity compared to bulk tumor cells. They also provide a broad overview of the use of murine models to investigate genetic reprogramming at the inception of Leukemia development. The concluding paragraph concerns new therapeutic strategies exploiting the processes characterizing the advent of Leukemia Initiating Cells.
Minor comments:
Point 1: - The authors should proofread the review, as there are a few minor typographical and grammar errors.
- The group could find a synonym for the phrase “in the same line of thought (or evidence)” which is often reposted at lines 157, 234, 288, 381, 489, 553
- Please check that acronyms are specified when first appearing in the paper. For instance, in line 176 Hematopoietic Stem and Progenitor Cells (HSPCs)
- In line 473 there is the URL of the paper “NUP98-PHF23 and NUP98-HOXD13 oncogenes confer aberrant self-renewal potential to thymocyte progenitors”. If the authors plan to use it as reference they have to add it as ref. 137 and then renumber all subsequent references.
- Authors should check that the symbols of genes and proteins are respectively in italics and in capital letters.
Response 1: We thank the reviewer to have noticed the redundancies. We found synonyms and changed the sentences, accordingly. We have also specified acronyms, ask the editors to add the reference, verified all the symbols of genes and proteins and proofread the review to find few typographical and grammar errors.
Point 2: - I would suggest, if possible, to add a couple of figures. For example:
- A figure regarding relapsed leukemic cells evolved from clones with genetic alterations and not from bulk cells
- A figure about the concept of self-renewal in leukemic initiation
- A figure that displays the role of bHLH and LMO1/2 in pre-LSCs activation
- A figure concerning the role of PAX5 in normal and abnormal B-cell development
Response 2: We agree with the reviewer that an illustration lacks in the manuscript. Therefore, we added a figure describing a model of leukemic evolution integrating the concepts of oncogene-induced reprogramming, self-renewal, leukemia initiation and transformation.
Reviewer 3 Report
It is an excellent and well-organized review. I would recommend to publish it in the present form. It would have been important to expand the last paragraph discussing the landscape of the current strategies for developing new cancer therapies. Targeting genes encoding for key players in controlling proliferation and differentiation in ALL, was a dream but unfortunately with very few examples of success in the clinic so far. A note of caution in that sense can be appropriate!
Author Response
Response to Reviewer 3 Comments
Comments and Suggestions for Authors
It is an excellent and well-organized review. I would recommend to publish it in the present form.
Point 1: It would have been important to expand the last paragraph discussing the landscape of the current strategies for developing new cancer therapies. Targeting genes encoding for key players in controlling proliferation and differentiation in ALL, was a dream but unfortunately with very few examples of success in the clinic so far. A note of caution in that sense can be appropriate!
Response 1: We really thank the reviewer to give us the opportunity to add this note of caution, that we did in the last paragraph and in the conclusion.